# Ion Acoustic Shocks in a Weakly Relativistic Ion-Beam Degenerate Magnetoplasma

**Rupinder Kaur and Nareshpal Singh Saini *** 

Department of Physics, Guru Nanak Dev University, Amritsar 143005, India; rupinderkaur.rk568@gmail.com
* Correspondence: nssaini@yahoo.com

**Abstract:** A theoretical investigation is carried out to study the propagation properties of ion acoustic shocks in a plasma comprising of positive inertial ions, weakly relativistic ion beam and trapped electrons in the presence of a quantizing magnetic field. By using the reductive perturbation technique, the Korteweg–de Vries-Burgers (KdVB) equation and oscillatory shocks solution are derived. The characteristics of such kinds of shock waves are examined and discussed in detail under suitable conditions for different physical parameters. The strength of the magnetic field, ion beam concentration and ion-beam streaming velocity have a great influence on the amplitude and width of the shock waves and oscillatory shocks. The results may be useful to study the characteristics of ion acoustic shock waves in dense astrophysical regions such as neutron stars.

**Keywords:** relativistic ion-beam; reductive perturbation method; KdVB equation; oscillatory shocks

## 1. Introduction

A number of investigations have confirmed that the presence of a high electron/ion beam in various space/astrophysical environments [1–3] makes drastic changes in the propagation properties of nonlinear structures (e.g., solitons, shocks, rogue waves, double layers etc.). During recent years, a variety of investigations have been reported to analyze the characteristic properties of different kinds of nonlinear structures in a multicomponent plasmas in the presence of an ion beam in the framework of Maxwellian/non-Maxwellian distributions. The basic reductive perturbation approach was used to investigate the nonlinear propagation of modest but finite amplitude dust ion-acoustic waves (DIAWs) in an ion-beam-driven plasma, including Boltzmann electrons, positive ions, and stationary negatively charged dust grains [4]. It was illustrated that the ion-beam component and other plasma parameters have a great influence on the amplitude as well as the width of DIAWs. The study of distinctive features of dust acoustic (DA) cnoidal waves in unmagnetized multicomponent dusty plasma penetrated by an ion beam was carried out theoretically [5]. The KdV equation was obtained to study the impact of different plasma parameters on the distinctive features of negative potential DA cnoidal waves using the reductive perturbation approach. Kaur et al. [6] reported the characteristics of compressive and rarefactive ion acoustic solitary waves (IASWs) in an unmagnetized plasma in the presence of an ion beam. The KdV, modified KdV, and Gardner equations were developed using the reductive perturbation technique to investigate the effect of an ion beam and other plasma factors on the properties of IASWs in various circumstances. The cylindrical/spherical Korteweg–de Vries equation was derived in a plasma comprised of superthermal electrons and ions and embedded with an ion beam in a nonplanar geometry to study the propagation characteristics of dust acoustic solitary and rouge waves [7]. Deka et al. [8] explored the nonlinear propagation of small amplitude ion acoustic solitary waves in a relativistic degenerate magnetoplasma in the presence of an ion beam. A plasma comprised of positive ions and a weakly relativistic ion beam in the presence of quantized trapped electrons was considered. The nonlinear equations

describing the development of a solitary waves were developed in terms of the Zakharov–Kuznetsov (Z-K) equation. A one-dimensional quantum hydrodynamic model was used to investigate the nonlinear propagation of ion acoustic waves in unmagnetized quantum plasma in the presence of an ion beam [9]. The reductive perturbation approach was used to generate the Korteweg–de Vries equation. The role of an ion beam in quantum plasma appeared to have a major impact on the formation and structure of solitary waves. Kaur et al. [10] developed the KdV-type equation to explore the propagation characteristics of ion acoustic (IA) cnoidal waves in a magnetized degenerate quantum plasma composed of positive inertial ions, weakly relativistic ion beam, and trapped electrons. The combined effects of temperature degeneracy, quantizing magnetic field, and other physical parameters on the properties of IA cnoidal waves were analyzed.

The importance of degenerate plasma and relativistic effects have been well described in different investigations with the focus of the study on the environment of astrophysical objects (namely viz., white dwarfs and neutron stars) [11–15]. The pseudo-potential approach was used to investigate arbitrary amplitude solitary waves (SWs) and double layers (DLs) in an ultra-relativistic degenerate dense dusty plasma (containing ultra-relativistic degenerate ultra-cold electron fluid, inertial ultra-cold ion fluid, and negatively charged static dust) by Mamun et al. [16]. In a dense relativistically degenerate plasma, Ghai et al. [17] investigated the connection between ion acoustic waves (IAWs) and a neutrino beam experiencing flavor oscillations. The effect of neutrino beam characteristics on the instability development rate was calculated numerically, including the energy of the incident neutrino beam and the eigenfrequency of neutrino flavor oscillations. The effects of spin-up and spin-down degenerate electrons on IA cnoidal waves in a magnetized degenerate quantum plasma were studied using a theoretical approach [18]. The solution for IA cnoidal waves was found using the Sagdeev pseudopotential technique, and the KdV type equation was constructed using the reductive perturbation method. From analysis, it was seen that only positive potential cnoidal waves are produced. The quantum hydrodynamic equations were used to examine the oblique propagation of quantum electrostatic solitary waves in a magnetized relativistic quantum plasma [19] by deriving the ZK equation. It was found that two types of quantum acoustic modes, a slow acoustic mode and a rapid acoustic mode, may propagate. El-Labany et al. [20] reported a paper on the interaction of two IA solitary waves in a magnetized relativistic degenerate plasma. After obtaining two Korteweg–de Vries (KdV) wave equations that describe the interacting IASs using the extended Poincaré–Lighthill–Kuo (PLK) technique, the phase changes owing to interaction were computed. In a degenerate quantum plasma, Saini et al. [21] developed the KdV equation to explore the properties of IA solitary waves under the impact of spin-up and spin-down electrons. The nonlinear Schrödinger equation (NLSE) was obtained from the KdV equation using suitable transformations. Freak waves were examined using the NLSE solutions of Akhmediev breathers and the Kuznetsov–Ma breathers.

A number of studies have witnessed the influence of trapped electrons in the regimes of laboratory and space plasmas [22–26]. Solitary and breather waves in a plasma system, including classical ions and degenerate trapped electrons, were studied using the reductive perturbation approach and the nonlinear Schrödinger equation [24]. In degenerate dense plasmas generated by ion beam, the influence of the quantized magnetic field and trapped electrons on the properties of ion acoustic solitary waves was investigated [27]. The development of the solitary wave in such plasmas was given by the Zakharov–Kuznetsov (Z–K) equation, which takes into account classical magnetized positive ions and ion beams, as well as the magnetically quantized trapped electrons. The impact of trapped electrons with a quantized magnetic field in a degenerate quantum plasma was used to study the nonlinear propagation of IA shock waves [25]. The reductive perturbation approach was used to develop the nonlinear KdVB equation, and the impact of various plasma parameters on IA shock waves was demonstrated. In quantum plasmas, a theoretical examination of nonlinear propagation of ion acoustic shock waves in the presence of trapping effects and Landau quantization was carried out [26]. The many KdVB solutions were examined,

and the excitation from KdV oscillations to the shock solution was explained in the presence of electron orbital motion. Landau quantization had a catastrophic influence on the height (strength) of the nonlinear shock structures.

During the five decades, the study on understanding the dynamics of nonlinear structures is one of the basic research topics in plasma physics. The mechanism of formation of shock structures has been illustrated in detail in an investigation reported by Kaur et al. [28]. The solutions of the Burgers/KdVB equation describe the formation of shock structures in the given plasma environment. The effect of different plasma parameters on structure and strength of shock waves was investigated. Ion-acoustic shock waves (IASWs) were investigated in a degenerate relativistic plasma nuclei of heavy elements by deriving the KdVB equation [29]. In a degenerate relativistic magneto-rotating quantum plasma, a theoretical examination of heavy nucleus acoustic (HNA) nonlinear structures was carried out [30]. The Zakharov–Kuznetsov–Burgers (ZKB) equation was developed using the reductive perturbation approach, and the impact of various plasma properties on the features of oscillatory shocks was investigated. In the recent past, Singh et al. [31] studied the influence of the anisotropy effect on electron acoustic shocks by deriving the KdVB equation in a superthermal magnetoplasma. Very recently, dressed shock waves (DSWs) have been investigated by Kaur et al. [28] in a degenerate quantum plasma made up of inertial heavy and light nuclei, as well as inertia-less ultra-relativistic degenerate electrons, due to the contribution of higher-order nonlinearity and dissipation effects. The higher-order effects of nonlinearity and dissipation were accounted for to derive an inhomogeneous Burgers-type equation. The influence of various plasma parameters has been analyzed on the characteristic properties of positive potential shocks and DSWs.

The most prominent theory for studying the dynamical properties of any plasma system is bifurcation theory. Samanata et al. [32] presented a study to demonstrate the features of nonlinear traveling wave solutions of the Kadomtsev–Petviashvili (KP) and Zakharov–Kuznetsov (ZK) equations by employing bifurcation theory. Using the reductive perturbation approach, Saha and Chatterjee [33] used the bifurcation theory to illustrate the prominent characteristics of nonlinear electron acoustic waves in an unmagnetized quantum plasma consisting of cool and hot electrons. In the present study, we have investigated the characteristic properties of shock waves in a plasma comprising of positive ion fluid, a weakly relativistic ion beam and trapped electrons in the presence of a quantizing magnetic field. By employing the reductive perturbation technique, the KdVB equation is derived, and further oscillatory solutions are studied. The manuscript is arranged as follows: Section 2 presents the fluid equations. In Section 3, the KdVB equation is obtained. In Section 4, the parametric analysis is provided, and in Section 5, the results are highlighted.

## 2. Basic Equations

The method proposed by Landau and Lifshitz [34] has been employed to derive the expressions for free electrons and trapped quantized degenerate electrons. Electrons moving in a plane perpendicular to the magnetic field become quantized in the presence of a strong magnetic field with energy levels $\epsilon_e^l = l\hbar\omega_{ce} + (p^2/2m_e) - e\phi$, where $-e\phi$ is the potential of the trapped electrons, $\omega_{ce} = eB_0/m_e c$ and $p_z$ is the momentum of the electrons along the direction of the magnetic field. Electron trapping occurs for $e\phi = l\hbar\omega_{ec} + (p_z^2/2m_e)$, where $\epsilon_e^l < 0$ and $\epsilon_e^l > 0$ are the energy of trapped and free electrons. The electrons are occupied within the range $\epsilon$ to $\epsilon + d\epsilon$, and summing up all Landau levels, the number density of electrons can be written as [34],

$$n_e = \frac{p_{Fe}^2 H}{2\pi^2\hbar^3}\sqrt{\frac{m_e}{2}}\sum_{l=0}^{\infty}\int_0^{\infty}\frac{\epsilon^{-1/2}}{1+exp(\epsilon - W/T)}d\epsilon, \tag{1}$$

where $W = e\phi + \mu - l\hbar\omega_{ce}$, $\mu$ is the chemical potential, $H = \hbar\omega_{ce}/\epsilon_{Fe}$, $n_{e0} = (p_{Fe}^3/3\pi^2\hbar^3)$ is the equilibrium density, $p_{Fe}$ is the momentum on the Fermi surface, $T = (\pi T/2\sqrt{2}\epsilon_{Fe})$.

The effect of the quantizing magnetic field appears through $H$. On solving integration on RHS of Equation (1), we obtain [10,27]

$$\frac{n_e}{n_{e0}} = \frac{3}{2}H(1+\phi)^{1/2} + (1+\phi-H)^{3/2} - \frac{HT^2}{2}(1+\phi)^{-3/2} + T^2(1+\phi-H)^{-1/2}, \quad (2)$$

where $\phi = e\phi/\epsilon_{Fe}$. On expanding R.H.S. of Equation (2) upto order of $\phi^2$, we obtain,

$$\frac{n_e}{n_{e0}} = \frac{H}{2}(3-T^2) + \frac{3H}{4}(1+T^2) - \frac{3H}{16}(1+5T^2) + \frac{3}{2}\left(\frac{H}{2}(1+T^2) + (1-H)^{1/2}\right.$$
$$\left. - \frac{T^2}{3}(1-H)^{-3/2}\right)\phi + \frac{3}{8}\left(\frac{-H}{2}(1+5T^2) + (1-H)^{-1/2} - T^2(1-H)^{-5/2}\right)\phi^2. \quad (3)$$

To investigate the nonlinear propagation properties of shock waves, we consider a plasma consisting of positive ions, weakly relativistic ion beam and trapped electrons in the presence of quantizing magnetic field. The basic fluid equations (continuity, momentum and Poisson) are given as [10,27],

$$\frac{\partial N_j}{\partial T} + \nabla(N_j U_j) = 0, \quad (4)$$

$$\left(\frac{\partial}{\partial T} + U_j\nabla\right)(\gamma_j U_j) = \left(\frac{e}{m_j}\right)E + \left(\frac{U_j}{c}\right) \times B + \frac{\mu}{m_j N_j}\nabla^2 U_j, \quad (5)$$

$$\nabla^2\Phi = 4\pi e(N_e - N_i - N_b), \quad (6)$$

where the relativistic factor is given by $\gamma_j = (1 - u_j^2/c^2)^{-1/2}$, $U_j^2 = U_{jX}^2 + U_{jY}^2 + U_{jZ}^2$, $(j = i, b)$ and $E = -\nabla\Phi$.

The component form of the normalized version of the continuity and momentum equation for ions and ion-beams is expressed as:

$$\frac{\partial n_j}{\partial t} + \frac{\partial(n_j u_{jx})}{\partial x} + \frac{\partial(n_j u_{jy})}{\partial y} + \frac{\partial(n_j u_{jz})}{\partial z} = 0, \quad (7)$$

$$\gamma_j\frac{\partial u_{jx}}{\partial t} + \left(u_{jx}\frac{\partial}{\partial x} + u_{jy}\frac{\partial}{\partial y} + u_{jz}\frac{\partial}{\partial z}\right)(\gamma_j u_{jx}) = -\rho_j\frac{\partial\phi}{\partial x} + \Omega_b u_{jy} + \eta\frac{\partial^2 u_{jx}}{\partial x^2}, \quad (8)$$

$$\gamma_j\frac{\partial u_{jy}}{\partial t} + \left(u_{jx}\frac{\partial}{\partial x} + u_{jy}\frac{\partial}{\partial y} + u_{jz}\frac{\partial}{\partial z}\right)(\gamma_j u_{jy}) = -\rho_j\frac{\partial\phi}{\partial y} - \Omega_b u_{jx} + \eta\frac{\partial^2 u_{jy}}{\partial y^2}, \quad (9)$$

$$\gamma_j\frac{\partial u_{jz}}{\partial t} + \left(u_{jx}\frac{\partial}{\partial x} + u_{jy}\frac{\partial}{\partial y} + u_{jz}\frac{\partial}{\partial z}\right)(\gamma_j u_{jz}) = -\rho_j\frac{\partial\phi}{\partial z} + \eta\frac{\partial^2 u_{jz}}{\partial z^2}, \quad (10)$$

The normalized form of Poisson's equation is:

$$\frac{\partial^2\phi}{\partial x^2} + \frac{\partial^2\phi}{\partial y^2} + \frac{\partial^2\phi}{\partial z^2} = \mu_e n_e - \mu_b n_b - n_i. \quad (11)$$

where $j = i, b$, $\gamma_j = (1 - v_j^2/c^2)^{-\frac{1}{2}}$, $(\gamma_i = 1)$, $\eta = \frac{\mu\omega_j}{m_j n_j c_s^2}$, $\mu_e = \frac{N_{e0}}{N_{i0}}$, $\mu_b = \frac{N_{b0}}{N_{i0}}$, $\rho_j = \frac{m_i}{m_j}$ and $\Omega_b = \frac{eB_0}{\omega m_j c_j}$. The scaling parameters used for the normalization of above equations are expressed as: $t = \omega_j T$, $x, y, z = (X, Y, Z)/\lambda_{Fe}$, $\phi = \left(\frac{\epsilon_{Fe}}{e}\right)\Phi$, $n_j = \frac{N_j}{N_{j0}}$, $u_j = \frac{U_j}{C_s}$ where $\lambda_{Fe}\omega_j = C_s$, $\omega_j = \left(\frac{4\pi n_{i0}e^2}{m_j}\right)^{1/2}$ and $C_s = \sqrt{\frac{\epsilon_{Fe}}{m_j}}$.

### 3. Derivation of the KdV–Burgers Equation

We have used the reductive perturbation method to derive the KdVB equation. The stretching coordinates are considered as:

$$\xi = \epsilon^{1/2}(l_x x + l_y y + l_z z - \lambda t) \quad and \quad \tau = \epsilon^{3/2} t, \tag{12}$$

where $\lambda$ is the phase velocity, and $\epsilon$ is the smallest parameter that describes the strength of nonlinearity. The state variables are expanded as:

$$n_j = 1 + \epsilon n_j^{(1)} + \epsilon^2 n_j^{(2)} + ...$$

$$u_{bx,y} = \epsilon^{3/2} u_{bx,y}^{(1)} + \epsilon^2 u_{bx,y}^{(2)} + ...$$

$$u_{bz} = V_{b0} + \epsilon u_{bz}^{(1)} + \epsilon^2 u_{bz}^{(2)} + ...$$

$$u_{ix,y} = \epsilon^{3/2} u_{ix,y}^{(1)} + \epsilon^2 u_{ix,y}^{(2)} + ...$$

$$u_{iz} = \epsilon u_{bz}^{(1)} + \epsilon u_{bz}^{(2)} + ...$$

$$\phi = \epsilon \phi^{(1)} + \epsilon^2 \phi^{(2)} + ... \tag{13}$$

We consider $\eta = \epsilon^{1/2}\eta_0$, where $\eta_0$ is the finite quantity of fluid viscosity. It is thought that the values of $\eta$ for heavier plasma fluids are expected to be less than unity in various experimental scenarios. The same scaling parameter $\epsilon$ can be used to scale the smallness of $\eta$ as for the wave amplitudes. Furthermore, such a scaling of $\eta$ is taken into account in such a way that it only affects the dissipative term and not the dispersive or nonlinear components. This is when the reductive perturbation approach comes into play. Otherwise, the fundamental concept for the dissipation source may not be applicable for wave dynamics. We substitute Equations (12) and (13) in Equations (7)–(11) and collect coefficients of different powers of $\epsilon$. The lowest order coefficients of $\epsilon$ for continuity equations, momentum equations and Poisson's equation after analytical manipulations yield the following equations in first-order quantities.

$$n_i^{(1)} = \frac{l_z^2}{\lambda^2}\phi^{(1)}, \tag{14}$$

$$u_{iz}^{(1)} = \frac{l_z}{\lambda}\phi^{(1)}, \tag{15}$$

$$n_b^{(1)} = \frac{l_z^2}{\gamma_{b1}(\lambda - V_{b0}l_z)^2}\phi^{(1)}, \tag{16}$$

$$u_{bz}^{(1)} = \frac{l_z}{\gamma_{b1}(\lambda - V_{b0}l_z)}\phi^{(1)}, \tag{17}$$

From the first order Equations (14)–(17), we have determined the dispersion relation as

$$\frac{l_z^2}{\lambda^2} = \mu_e \alpha_1 + \frac{\mu_b l_z^2}{\gamma_{b1}(\lambda - V_{b0}l_z)^2}, \tag{18}$$

where $\alpha_1 = \frac{3}{2}\left(\frac{H}{2}(1 + T^2) + (1 - H)^{1/2} - \frac{T^2}{3}(1 - H)^{-3/2}\right)$. This is a bi-quadratic equation and it is not exactly solvable. We solved this equation numerically using MATHEMATICA-

10 and obtained one pair of imaginary roots, one negative and one positive root. Since with the negative root, no solitary structures are formed, the positive root of Equation (18) (i.e., positive value of $\lambda$) has been used for further analysis. Neglecting the relativistic effects in the present investigation, the dispersion relation Equation (18) of this investigation matches with the dispersion relation in the investigation of Deka and Dev [27]. Further collecting higher-order coefficients of $\epsilon$ (see Appendix A—Equations (A1)–(A5)) and eliminating higher-order terms by making use of Equations (14)–(17) in Equations (A1)–(A5), we obtain the following KdVB equation

$$\frac{\partial \phi}{\partial \tau} + A\phi\frac{\partial \phi}{\partial \xi} + B\frac{\partial^3 \phi}{\partial \xi^3} = C\frac{\partial^2 \phi}{\partial \xi^2},$$ (19)

where,

$$A = \frac{p}{q}, \quad B = \frac{r}{q} \quad and \quad C = \frac{-\eta_0}{2}$$

with

$$p = \left[ \frac{l_z^3 \gamma_{b2} \mu_b}{2\gamma_{b1}(\lambda - V_{b0}l_z)^2} - \frac{3l_z^4 \mu_b}{2\gamma_{b1}^2(\lambda - V_{b0}l_z)^4} + \frac{3l_z^4}{2\lambda^4} - \frac{\mu_e \alpha_2}{2} \right],$$

$$r = \left[ 1 + \frac{\lambda(1 - l_z^2)\mu_b}{\Omega_{Bb}^2} + \frac{(1 - l_z^2)}{\Omega_{bi}^2} \right],$$

$$q = 2\left[ \frac{l_z^2}{\lambda^3} + \frac{\mu_b l_z^2}{\gamma_{b1}(\lambda - V_{b0}l_z^3)} \right].$$ (20)

For mathematical simplicity, we have considered here $\phi^{(1)} = \phi$ in Equation (19). The solution of Equation (19) describes the propagation of shock waves, where A is the nonlinear coefficient that determines the polarity of the shock structures, B is the dispersion coefficient and C is the dissipation coefficient. In the absence of dissipation effects, the KdVB Equation (19) is transferred to the KdV equation. In the absence of relativistic effects, this KdV equation agrees with the KdV equation obtained in the limiting case (when $C = 0$) from the ZK Equation (17) derived by Deka and Dev [27]. Further, it is highlighted that the nonlinear structures observed in the present investigation are shocks, whereas in Reference [27], nonlinear structures are solitons.

### 3.1. Solution of the KdV–Burgers Equation

To find the stationary solution of Equation (19), another frame of reference is considered as $\chi = (\xi - U\tau)$ where $U$ is the speed of shocks. By introducing single variable transformations in Equation (19) and integrating, we obtain the following equation:

$$-U\phi + \frac{A}{2}\phi^2 + B\frac{d^2\phi}{d\chi^2} = C\frac{d\phi}{d\chi}.$$ (21)

The above second-order differential equation can be solved by using the tanh method. After using the tanh approach, the analytical stationary solution of Equation (19) is determined as [35,36],

$$\phi_{shock} = \phi_m\left( 4 - \left[ 1 + tanh\left(\frac{\chi}{W}\right) \right]^2 \right),$$ (22)

here, $\phi_m = \frac{3C^2}{25AB}$ is the peak amplitude and $W = \frac{10B}{C}$ is the width and the speed of shock waves is $U = \frac{6C^2}{25B}$. In a limiting case, when the dissipation coefficient vanishes ($C \to 0$), Equation (19) becomes KdV equation, and its solution is

$$\phi_{soliton} = \phi_0 sech^2 \left[ \frac{\chi}{W} \right], \tag{23}$$

where $\phi_0 = \frac{3U_0}{A}$ is the peak amplitude of the solitary wave, and $W = \sqrt{\frac{4B}{U_0}}$ is the width of the solitary wave. The influence of different plasma parameters on the characteristics of IA solitons ($\mu_b$, $V_{b0}$, $\Omega_b$, $H$ and $T$) has been analyzed.

### 3.2. Solution of Oscillatory Shocks

Now, we are going to check another type of solution of Equation (19) by dealing with certain asymptotic boundary conditions. From KdVB Equation (19), we write,

$$\frac{1}{B} \frac{\partial \phi}{\partial \tau} + \frac{A}{B} \phi \frac{\partial \phi}{\partial \xi} + \frac{\partial^3 \phi}{\partial \xi^3} - \frac{C}{B} \frac{\partial^2 \phi}{\partial \xi^2} = 0. \tag{24}$$

Using transformation $\chi = (\xi - U\tau)$ in the above equation and supposing that $\phi = \phi_0 + \Phi$, where $\phi_0 = 2U/A$, $\phi \ll \phi_0$ and on linearizing Equation (24) with respect to $\phi$ we obtain,

$$\frac{d^2\Phi}{d\chi^2} - \frac{C}{B} \frac{d\Phi}{d\chi} - \frac{U}{B} \Phi = 0, \tag{25}$$

The above equation represents a well-known solution of a damped harmonic oscillator. The oscillatory shock wave solution for Equation (19) is given by [37],

$$\phi_{osc} = \frac{2U}{A} + Q \exp(-\beta\chi) \cos(\omega_1 \chi), \tag{26}$$

where $\beta = \frac{-C}{2B}$ is the damping factor, $\omega_1 = \sqrt{\frac{U}{B} \left( 1 - \frac{C^2}{4UB} \right)}$, and $Q$ is the arbitrary constant. $\frac{U}{B}$ represents natural frequency of the system.

### 3.3. Bifurcation Analysis

The Burger term of Equation (21) describes the homogenous and dissipative weakly relativistic degenerate magnetoplasma, and the phase shift can be solved by $\frac{dP}{d\chi}$ rather than $P$. Equation (21) is rearranged as

$$\frac{d^2\phi}{d\chi^2} + h\left( \phi, \frac{d\phi}{d\chi} \right) \frac{d\phi}{d\chi} + G(\phi) = 0, \tag{27}$$

where $h$ and $G$ are obtained by comparing Equations (21) and (27). For the conservative case ($h = 0$), the total energy is given as

$$P = \frac{1}{2} \left( \frac{d\phi}{d\chi} \right)^2 + V(\phi) \quad and \quad \frac{dP}{d\chi} = \frac{d\phi}{d\chi} \left( \frac{d^2\phi}{d\chi^2} + \frac{dV}{d\phi} \right). \tag{28}$$

In Equation (27), $G(\phi) = \frac{dV}{d\phi}$ and the total derivative of $P$ is obtained from

$$\frac{dP}{d\chi} = -h\left( \phi, \frac{d\phi}{d\chi} \right) \left( \frac{d\phi}{d\chi} \right)^2. \tag{29}$$

It reduces with $\chi$ if $h > 0$. The term $\frac{dP}{d\chi}$ leads to the KdV Burgers equation,

$$\frac{dP}{d\chi} = -\frac{C}{B}\left(\frac{d\phi}{d\chi}\right)^2. \tag{30}$$

Equation (30) describes the nonconservative case. It is also a decreasing function. If $C = 0$, then Equation (21) leads to a conservative system,

$$P = \frac{1}{2}\left(\frac{d\phi}{d\chi}\right)^2 + V(\phi), \tag{31}$$

where $V(\phi) = \frac{A}{6B}\phi^3 - \frac{U}{2B}\phi^2$. For $A > 0$ and $B > 0$, the bifurcation analysis can be studied by using Equation (31). The existence condition for solitons must be satisfied, i.e., $\frac{d^2V}{d\phi^2} < 0$. From the phase plots, one can obtain the periodic orbits at the saddle point that defines the family of periodic wave solutions.

## 4. Parametric Analysis

We have considered a plasma consisting of positive ions, a weakly relativistic ion beam and trapped degenerate electrons in the presence of a quantizing magnetic field. The effects of a quantizing magnetic field and temperature are described by $H$ and $T$, respectively. The plasma density with range of the order $10^{26}$–$10^{28}$ cm$^{-3}$, a magnetic field with range $10^{9-11}$ G and a Fermi temperature of order $3.6277 \times 10^7$ K have been used to carry out numerical calculations in the present investigation [38,39].

It is analyzed numerically that the nonlinear coefficient ($A$) is reduced with the increase in the value of the ion beam density ratio ($\mu_b$). Further, it is observed numerically that the nonlinear coefficient ($A$) is increased with the increase in the value of the direction of the cosine ($l_z$). Since the nonlinear coefficient (A) is positive, only positive potential shocks are observed in this plasma system.

### 4.1. Variation of Shock Wave Profile

Figure 1a,b represents the variation of pulse profile of shocks for the different values of relativistic factor (via $\gamma_b$) and ion beam velocity (via $V_{b0}$). It is seen that the amplitude of shocks is decreased with an increase in the relativistic factor $\gamma_b$. This is due to the fact that the maximum amplitude of the shock wave profile is inversely proportional to the nonlinear coefficient ($A$). Therefore, with the increase in the relativistic factor (via $\gamma_b$), the nonlinear coefficient is increased, which in turn reduces the maximum amplitude of the shock wave profile. On the other hand, in Figure 1b, with an increase in the beam velocity ($V_{b0}$), the nonlinear coefficient ($A$) decreases, and due to a decrease in the nonlinear coefficient, the amplitude of the shock waves is increased. Figure 2a,b depicts the variation of the 3D shock wave profile for the different values of ion density ratio ($\mu_b$) and magnetic field strength ($H$). Figure 2a illustrates that as the value of ion beam density (via $\mu_b$) increases, the amplitude of the shock wave profile is increased. On the other hand, from Figure 2b, it is seen that with the increase in the magnetic field strength (H), the amplitude of the shock waves is decreased. Figure 3a presents the variation of the 3D shock profile for different values of temperature degeneracy ($T$). It is observed that as the value of $T$ increases, the amplitude of shock waves is reduced. Figure 3b depicts the variation of the 3D profile of shock waves for the different values of viscosity $\eta$. An increase in the value of $\eta$ allows the amplitude of the shock wave to flourish.

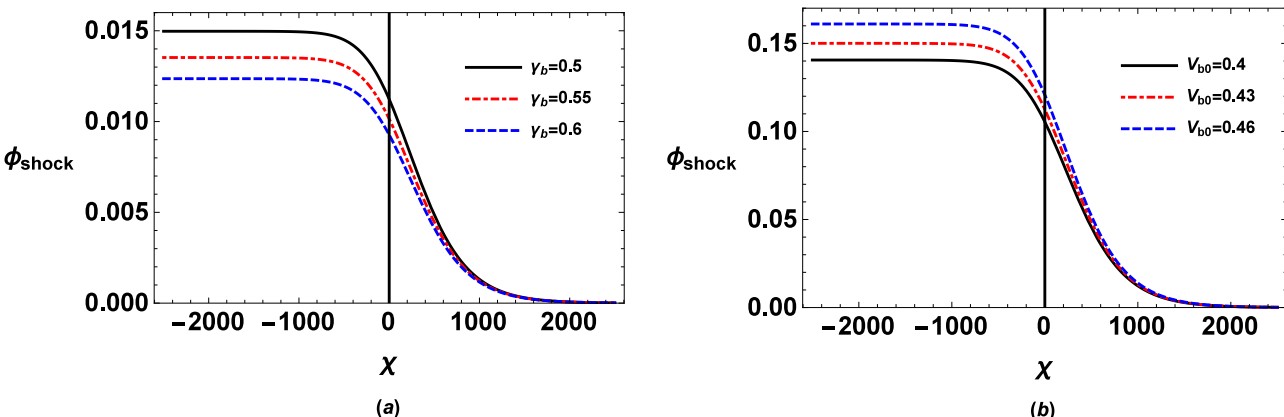

**Figure 1.** (Color online) Variation of the shock pulse profile for different values of (**a**) relativistic factor ($\gamma_b$), (**b**) beam velocity ($V_{b0}$), with fixed values of $l_z = 0.6$, $\eta = 0.4$, $\mu_b = 0.2$ $\gamma_b = 0.5$, $H = 0.2$ and $T = 0.2$.

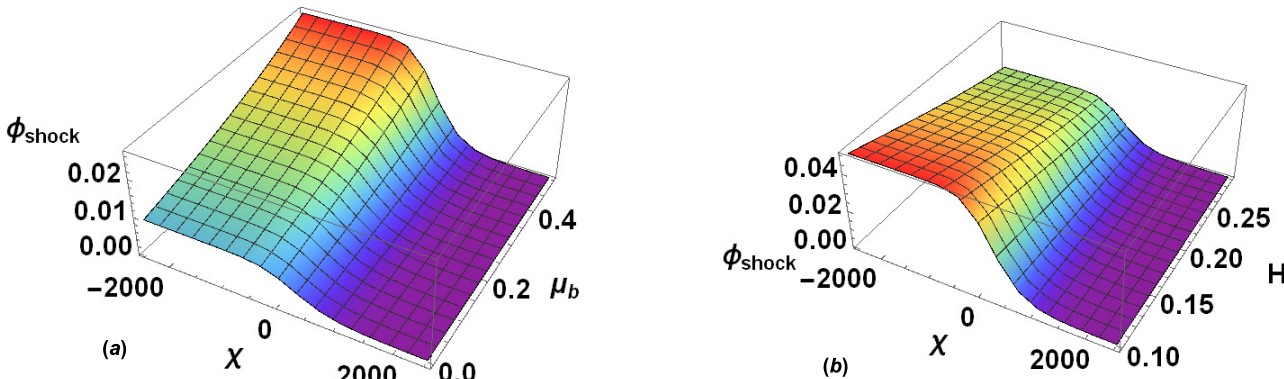

**Figure 2.** (Color online) Variation of the three-dimensional profile of the shock wave for different values of (**a**) ion-beam density ratio ($\mu_b$), (**b**) magnetic field strength ($H$), with fixed values of $l_z = 0.6$, $\eta = 0.4$, $V_{b0} = 0.4$, $\gamma_b = 0.5$, and $T = 0.2$.

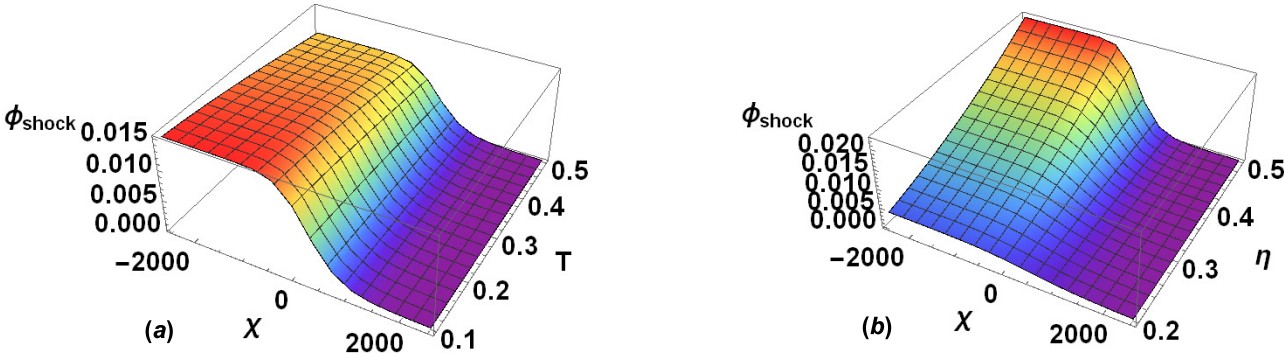

**Figure 3.** (Color online) Variation of Three-dimensional profile of shock wave for different values of (**a**) temperature degeneracy ($T$) (**b**) viscosity ($\eta$), with fixed values of $l_z = 0.6$, $\mu_b = 0.2$, $V_{b0} = 0.4$, $\gamma_b = 0.5$, and $H = 0.2$.

### 4.2. Variation of Oscillatory Shocks

The solution of the oscillatory shock waves can be presented in Equation (26), which is plotted against $\chi$ for different parameters. From Figure 4, it can be seen that the amplitude of oscillatory shocks is increased with the increase in $\mu_b$ (see Figure 4). Figure 5 depicts the influence of $\eta$ on the amplitude of the oscillatory shock waves. With the increase in $\eta$, the amplitude of the oscillatory shock waves is also increased. Figure 6 presents the variation of oscillatory shock waves for different values of the relativistic factor ($\gamma_b$). Figure 6a depicts the relativistic case, and Figure 6b depicts the nonrelativistic case. On comparing

the results in both Figure 6a,b it is observed that the amplitude of oscillatory shock waves for the nonrelativistic case is much larger than the relativistic case.

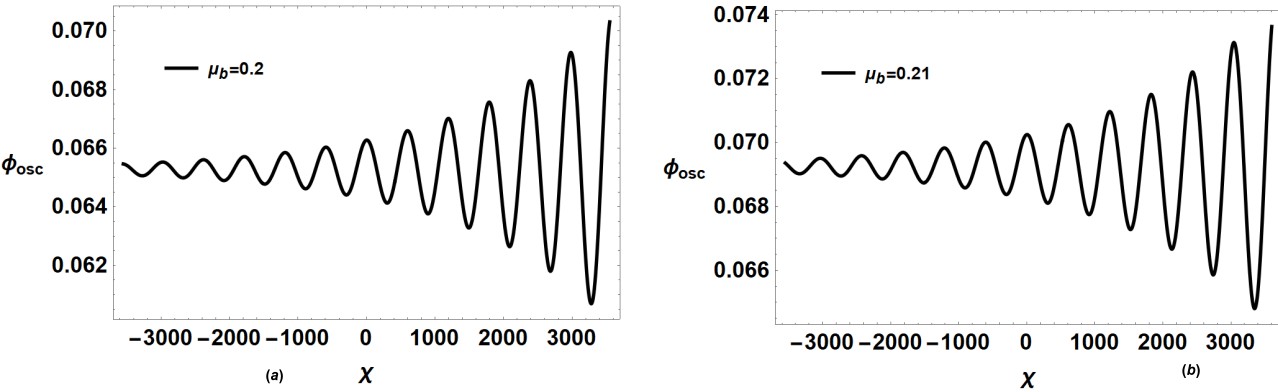

**Figure 4.** (Color online) Variation of oscillatory shock waves for different values of ion-beam density ratio ($\mu_b$) (**a**) $\mu_b = 0.2$ (**b**) $\mu_b = 0.21$, with fixed values of $l_z = 0.6$, $\eta = 0.05$, $V_{b0} = 0.4$, $\gamma_b = 0.5$, $H = 0.2$ and $T = 0.2$.

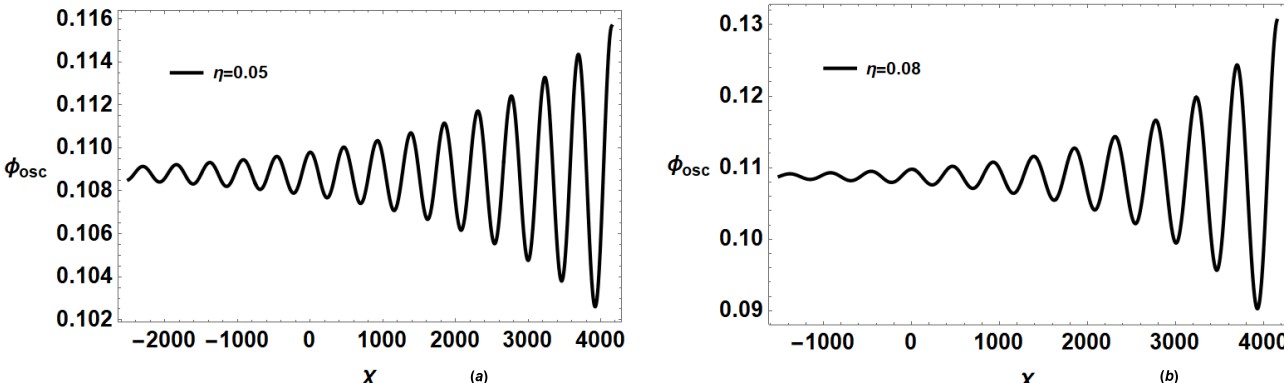

**Figure 5.** (Color online) Variation of oscillatory shock waves for different values of viscosity ($\eta$) (**a**) $\eta = 0.05$ (**b**) $\eta = 0.08$, with fixed values of $l_z = 0.6$, $\mu_b = 0.2$, $V_{b0} = 0.4$, $\gamma_b = 0.5$, $H = 0.2$ and $T = 0.2$.

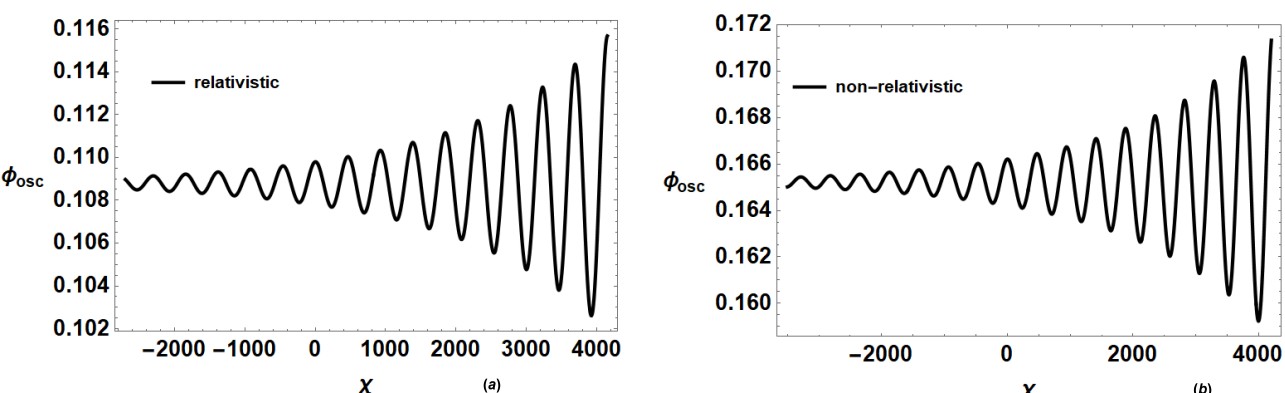

**Figure 6.** (Color online) Variation of oscillatory shock waves for different values of relativistic factor ($\gamma_b$) (**a**) for the relativistic case ($\gamma_b = 0.2$) (**b**) for the nonrelativistic case ($\gamma_b = 0$), with fixed values of $l_z = 0.6$, $\eta = 0.05$, $V_{b0} = 0.4$, $\mu_b = 0.2$, $H = 0.2$ and $T = 0.2$.

### 4.3. Variation of Sagdeev Potential and Phase Portraits

Equation (31) is used to study the bifurcation analysis. Figure 7a illustrates the variation of Sagdeev potential $G(\phi)$ vs. $\phi$ for different values of ion-beam density ($\mu_b$), magnetic field strength ($H$) and temperature degeneracy ($T$). As it is seen from Figure, the Sagdeev potential has one hump and one bend. It is observed that the depth of the Sagdeev potential decreases with an increase in $\mu_b$ and $H$. On the other hand, the depth of the Sagdeev potential increases with an increase in $T$. Figure 7b depicts a family of periodic orbits at $(2G/A, 0)$ comparable to the periodic wave solutions. The saddle point $(0,0)$ represents the solitary wave solution. $Q < 0$ shows that the family of periodic orbits around the center point at $(2G/A, 0)$ and $Q = 0$ represents the solitary wave solution (homoclinic orbit) at the saddle point (0,0). The case of $Q < 0$ corresponds to the breaking waves. Figure 7b shows a chain of open orbits that corresponds to a train of breaking of waves. From the bifurcation analysis, we have analyzed the characteristics of solitary and periodic waves for the different cases of $Q$. In studying the bifurcation analysis of waves, the phase portraits and planar dynamical systems have a great significance [32,33].

### 4.4. Variation of Solitary Wave Profile

In a limiting case, when the dissipation coefficient $C = 0$, the KdVB equation then approaches the KdV equation. Using the analytical solution of the KdV equation, we have also studied the formation of solitary waves. Figure 8a depicts the variation of a solitary wave profile for the different values of beam velocity $V_{b0}$. As the value of $V_{b0}$ increases, the amplitude of solitary waves increases and the width remains the same. Nonlinearity plays a very important role in the increase in the height of the solitary wave profile with an increase in the value of the beam velocity $V_{bo}$. Figure 8b presents the variation of the solitary wave profile for the different values of direction cosines $l_z$. It is observed that with the increase in $l_z$, the amplitude as well as the width of the solitary wave decreased. It is emphasized that different physical parameters have a great influence on the characteristics of shocks and solitary structures in a given plasma system.

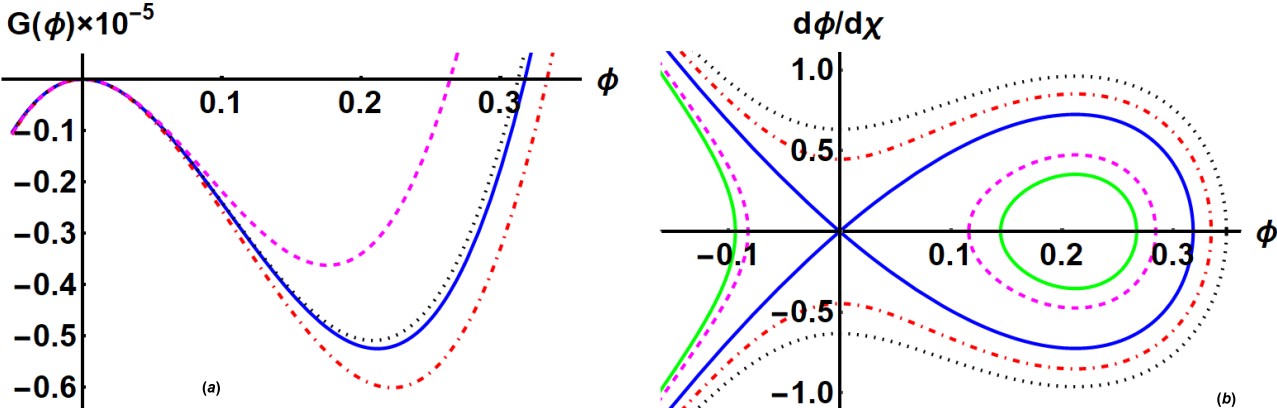

**Figure 7.** (Color online) (**a**) The variation of Sagdeev potential $(G(\phi))$ vs. $\phi$ with reference curve (blue solid curve) at $\mu_b = 0.2$, $V_{b0} = 0.4$, $H = 0.2$ and $T = 0.2$; black (dotted) curve for $\mu_b = 0.24$; red (dot-dashed) curve for $H = 0.4$; magenta (dashed) curve for $T = 0.4$. (**b**) The case of $Q < 0$ corresponds to the family of orbits (magenta dashed and green solid curves), $Q = 0$ represents the solitary wave (blue solid curve) and for $H > 0$, represents the chain of open orbits with a train of the breaking wave solution (red dot-dashed and black dotted curves).

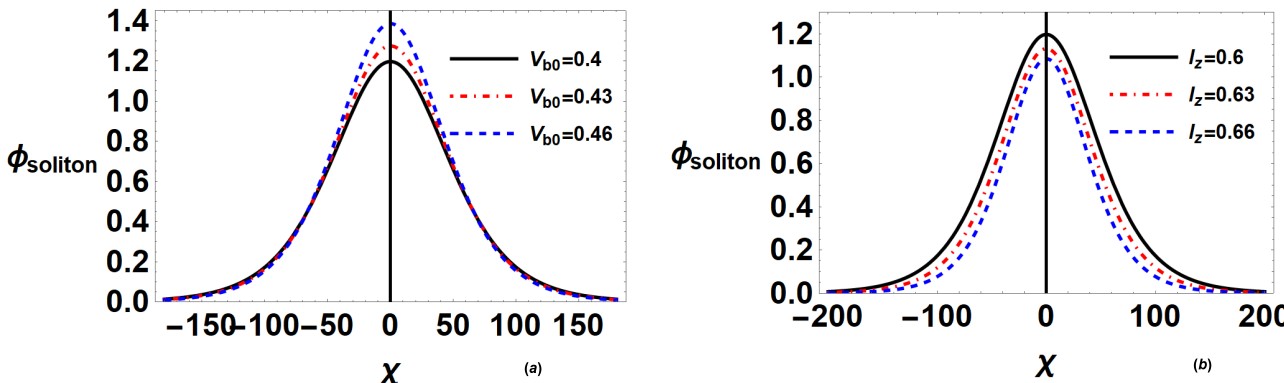

**Figure 8.** (Color online) Variation of solitary profile for different values of (**a**) beam velocity ($V_{b0}$), (**b**) direction cosine ($l_z$), with fixed values of $\mu_b = 0.2$, $U_0 = 0.02$, $H = 0.2$ and $T = 0.2$.

## 5. Conclusions

We have studied the characteristic properties of shock waves and oscillatory shocks under the influence of different plasma parameters in a relativistic degenerate plasma with positive ions and a weakly relativistic ion-beam in the presence of magnetically quantized trapped electrons. Only positive potential shock waves are formed. An ion-beam plays an important role in the formation of different types of shocks and changes in their characteristics. The nonlinearity of the system decreases as the ratio of the concentration of ion-beam to the concentration of ion ($\mu_b$) and beam velocity $V_{b0}$ increases. The amplitude of the shock profile is decreased (increased) for the increase in the values of the relativistic factor $\gamma_b$ (beam velocity ($V_{b0}$)). Furthermore, with the increase in $\mu_b$ ($H$), the amplitude of the shock wave is increased (decreased). The influence of different plasma parameters on the oscillatory shock waves has also been studied. The amplitude of oscillatory shocks is increased with the increase in $\mu_b$ and $\eta$. The relativistic factor $\gamma_b$ has a strong influence on the amplitude of oscillatory shock waves. As we are moving from a relativistic to nonrelativistic case, $\gamma_b = 0$, the amplitude of oscillatory shock waves is enhanced. In the absence of dissipation, a bifurcation analysis is also presented, and the characteristics of solitary and periodic waves are studied. In a limiting case (with $C = 0$), the KdVB equation approaches the KdV equation. The amplitude of solitary waves is enhanced with an increase in the value of beam velocity ($V_{b0}$). This investigation may be useful to understand the nonlinear phenomena responsible for the formation of ion acoustic shock waves in astrophysical dense plasma environments, especially in white dwarfs.

**Author Contributions:** Conceptualization and methodology; and numerical analysis; writing—review and editing, R.K. and N.S.S. Both authors have read and agreed to the published version of the manuscript.

**Funding:** This research received no external funding.

**Institutional Review Board Statement:** Not applicable.

**Informed Consent Statement:** Not applicable.

**Data Availability Statement:** Not applicable.

**Acknowledgments:** Rupinder Kaur thanks the Department of Science and Technology (DST) for financial assistance via the DST-Purse initiative. The authors appreciate the Department of Science and Technology, Government of India, New Delhi, for funding this research effort under the DST-SERB project no. CRG/2019/003988.

**Conflicts of Interest:** The authors declare no conflict of interest.

## Appendix A

Comparing coefficients of higher-order terms of $\epsilon$ in the continuity equation for ion and ion beam, respectively, are,

$$-\lambda \frac{\partial n_i^{(1)}}{\partial \xi} + \frac{\partial n_i^{(1)}}{\partial \tau} + l_x \frac{\partial u_{ix}^{(2)}}{\partial \xi} + l_y \frac{\partial u_{iy}^{(2)}}{\partial \xi} + l_z \frac{\partial u_{iz}^{(2)}}{\partial \xi} + l_z \frac{\partial n_i^{(1)} u_{iz}^{(1)}}{\partial \xi} = 0, \qquad (A1)$$

$$-\lambda \frac{\partial n_b^{(1)}}{\partial \xi} + \frac{\partial n_b^{(1)}}{\partial \tau} + l_x \frac{\partial u_{bx}^{(2)}}{\partial \xi} + l_y \frac{\partial u_{by}^{(2)}}{\partial \xi} + l_z \frac{\partial u_{bz}^{(2)}}{\partial \xi} + l_z \frac{\partial n_b^{(1)} u_{bz}^{(1)}}{\partial \xi} + l_z V_{b0} \frac{\partial n_b^{(2)}}{\partial \xi} = 0, \qquad (A2)$$

Collecting next order terms of the momentum equation of the z-component for ions and beam,

$$\frac{\partial u_{iz}^{(1)}}{\partial \tau} - \lambda \frac{\partial u_{iz}^{(2)}}{\partial \xi} + l_z u_{iz}^{(1)} \frac{\partial u_{iz}^{(1)}}{\partial \xi} = -\frac{\partial \phi^{(2)}}{\partial \xi} + \eta_0 \frac{\partial^2 u_{iz}^{(1)}}{\partial \xi^2}, \qquad (A3)$$

$$\frac{\partial u_{bz}^{(1)}}{\partial \tau} + \lambda \gamma_{b1} l_z \frac{\partial u_{bz}^{(2)}}{\partial \xi} + \gamma_{b1} l_z u_{bz}^{(1)} \frac{\partial u_{bz}^{(1)}}{\partial \xi} - \lambda \gamma_{b1} \frac{\partial u_{bz}^{(2)}}{\partial \xi} = -l_z \frac{\partial \phi^{(2)}}{\partial \xi} + \eta_0 \frac{\partial^2 u_{iz}^{(1)}}{\partial \xi^2}, \quad (A4)$$

Collecting higher-order terms of Poisson's equation,

$$\frac{\partial^2 \phi^{(1)}}{\partial \xi^2} = \mu_e \alpha_1 \phi^{(2)} + \mu_e \alpha_2 \phi^{(1)} + \mu_b n_b^{(2)} - n_i^{(2)}. \qquad (A5)$$

where $\alpha_2 = \frac{3}{8}\left\{\frac{-H}{2}(1 + 5T^2) + (1 - H)^{-1/2} - T^2(1 - H)^{-5/2}\right\}$.

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
