# Peer review of "Ion Acoustic Shocks in a Weakly Relativistic Ion-Beam Degenerate Magnetoplasma"

_galaxies, doi:10.3390/galaxies9030064_

Round 1
Reviewer 1 Report
The authors have considered a beam-plasma model containing trapping and quantization effects. The reductive perturbation technique as well as the phase portrait are used to study the model. However, I see that the linear dispersion relation needs more analysis. Based on the latter many physics may be released. The authors make a fine survey on the topics though there are some recent related articles that could be cited rather than those belong to their research co-workers.
Author Response
C1: The authors have considered a beam-plasma model containing trapping and quantization effects. The reductive perturbation technique as well as the phase portrait are used to study the model. However, I see that the linear dispersion relation needs more analysis. Based on the latter many physics may be released. The authors make a fine survey on the topics though there are some recent related articles that could be cited rather than those belong to their research co-workers
Response: We have elaborated the linear analysis as per suggestions of the reviewer. The necessary references have been included in revised version of MS.
Reviewer 2 Report
Rupinder Kaur, et al. investigated the ion-acoustic shocks in a weakly relativistic ion-beam degenerate magnetoplasma. Basically, this is a good study supported by clear evidence and data, and a clear logic governing the whole study. I just have some suggestions for a minor revision.
- Please enhance the discussion on the background and the uniqueness of this work. Please introduce more references over the past 5 years.
- Please discuss more in terms of mechanism in the discussion part particularly in terms of physics.
Author Response
C1: Rupinder Kaur, et al. investigated the ion-acoustic shocks in a weakly relativistic ion-beam degenerate magnetoplasma. Basically, this is a good study supported by clear evidence and data, and a clear logic governing the whole study. I just have some suggestions for a minor revision.
- Please enhance the discussion on the background and the uniqueness of this work. Please introduce more references over the past 5 years.
Response: As per reviewer’s suggestions, we have made the necessary changes in the revised MS and highlighted the uniqueness of this work.
2. Please discuss more in terms of mechanism in the discussion part particularly in terms of physics.
Response: The necessary discussion has been included in the revised version of manuscript.
Reviewer 3 Report
It would be the case to introduce an analysis of the difference of the results of this paper with the methods and results of analogous papers about the same topic.
Author Response
C1: It would be the case to introduce an analysis of the difference of the results of this paper with the methods and results of analogous papers about the same topic.
Response: As per comments of the reviewer, we have compared our results with the findings of other similar plasma model.